# The Tubulin Code in Mitosis and Cancer

**DOI:** 10.3390/cells9112356

**Published:** 2020-10-26

**Authors:** Danilo Lopes, Helder Maiato

**Affiliations:** 1Chromosome Instability & Dynamics Group, i3S—Instituto de Investigação e Inovação em Saúde, Universidade do Porto, Rua Alfredo Allen 208, 4200-135 Porto, Portugal; danilo.lopes@i3s.up.pt; 2Instituto de Biologia Molecular e Celular, Universidade do Porto, Rua Alfredo Allen 208, 4200-135 Porto, Portugal; 3Cell Division Group, Experimental Biology Unit, Department of Biomedicine, Faculdade de Medicina, Universidade do Porto, Alameda Prof. Hernâni Monteiro, 4200-319 Porto, Portugal

**Keywords:** cancer, chromosomal instability, microtubule, mitosis, tubulin code, tubulin post-translational modifications

## Abstract

The “tubulin code” combines different α/β-tubulin isotypes with several post-translational modifications (PTMs) to generate microtubule diversity in cells. During cell division, specific microtubule populations in the mitotic spindle are differentially modified, but only recently, the functional significance of the tubulin code, with particular emphasis on the role specified by tubulin PTMs, started to be elucidated. This is the case of α-tubulin detyrosination, which was shown to guide chromosomes during congression to the metaphase plate and allow the discrimination of mitotic errors, whose correction is required to prevent chromosomal instability—a hallmark of human cancers implicated in tumor evolution and metastasis. Although alterations in the expression of certain tubulin isotypes and associated PTMs have been reported in human cancers, it remains unclear whether and how the tubulin code has any functional implications for cancer cell properties. Here, we review the role of the tubulin code in chromosome segregation during mitosis and how it impacts cancer cell properties. In this context, we discuss the existence of an emerging “cancer tubulin code” and the respective implications for diagnostic, prognostic and therapeutic purposes.

## 1. The Tubulin Code

Microtubules are dynamic, hollow cylindrical structures typically formed by thirteen laterally associated protofilaments of α/β-tubulin heterodimers that interact head-to-tail [1]. α- and β-tubulin proteins are encoded by several different genes (also known as tubulin isotypes) that diverge in their C-terminal tail regarding length and amino acid composition [2,3]. In eukaryotes, the expression and distribution of different tubulin isotypes is cell- and tissue-specific [2]. In addition, α- and β-tubulin isotypes may undergo multiple post-translational modifications (PTMs). As α/β-tubulin heterodimers polymerize into microtubules, the combination of isotype expression with PTMs generate microtubule diversity or a “tubulin code” (Figure 1), which has been implicated in the regulation of microtubule properties and functions underlying fundamental cellular processes [4,5].

Acetylation, detyrosination, polyglutamylation and polyglycylation are amongst the best characterized tubulin PTMs (Figure 1). Acetylation occurs in both α- and β-tubulins, more specifically at the luminal-side Lysine-40 (K40) of α-tubulin [6,7] and Lysine 252 (K252) of β-tubulin [8]. While K252 is modified by the acetyltransferase San [8], K40 is acetylated by the acetyltransferase MEC-17/αTAT1 [9,10] and deacetylated by histone deacetylase 6 (HDAC6) and sirtuin2 (SIRT2) [11,12]. When incorporated into microtubules, α-tubulin can also be detyrosinated, which consists on the catalytic removal of the last tyrosine present at the C-terminal tail of most isoforms by tubulin carboxypeptidases (TCPs), including the recently identified Vasohibin 1 (VASH1) and Vasohibin 2 (VASH2) complexes with their associated Small Vasohibin-Binding Protein (SVBP) [13,14,15,16,17,18,19]. As microtubules depolymerize, soluble detyrosinated α-tubulin can be retyrosinated by a highly specific tubulin tyrosine ligase (TTL) that closes the cycle [20,21]. Noteworthy, additional TCPs remain to be identified, as substantial α-tubulin detyrosination still occurs in human cells in which both Vasohibin-encoding genes were knocked out by CRISPR-Cas9 [14]. After detyrosination, α-tubulin C-terminal tails may also be subject to the removal of the penultimate and antepenultimate glutamates by cytosolic carboxypeptidases (CCPs) [22,23], leading to formation of the non-tyrosinatable Δ2- and Δ3-tubulin, respectively [24,25]. Additionally, C-terminal tails of both α- and β-tubulins undergo side-chain polyglutamylation and polyglycylation [26,27]. The single or consecutive addition of glutamate residues to the γ-carboxyl group of C-terminal tails is performed by several TTL-like (TTLL) (poly)glutamylases [5,28,29] and is/are removed by a set of CCPs known as deglutamylases [5,22,23,30]. Similarly, the addition of glycine residues relies on the (poly)glycylases TTLL3, TTLL8 and TTLL10 [31,32], but the identity of tubulin deglycylases remains unknown. Lastly, several other tubulin PTMs, such as methylation, polyamination, phosphorylation, ubiquitinylation, sumoylation, palmitoylation (reviewed in [5]) and O-GlcNAcylation [33] occur in the tubulin core structure adjacent to the C-terminal tails. These PTMs remain poorly characterized at the functional level but are likely to be implicated in microtubule assembly and dynamics [5,34,35].

## 2. The Tubulin Code in Mitosis

Mitosis relies on the critical contribution of microtubules, as well as several microtubule-associated proteins (MAPs) and motors, to regulate several key mechanisms underlying the faithful segregation of the genetic material during cell division. It involves the assembly of a specialized microtubule-based structure known as the mitotic spindle. Due to their intrinsic dynamic nature, mitotic spindle microtubules are vastly tyrosinated, i.e., remain essentially nonmodified (note that most gene-encoded α-tubulin isoforms carry a last Tyrosine residue at their C-terminal tails; see Figure 1). As some spindle microtubules become gradually stabilized due to the establishment of chromosome attachments at the kinetochore, as well as possible interactions between some interpolar microtubules, they become increasingly detyrosinated [19,36,37,38,39,40] (Figure 2). Likewise, kinetochore microtubules are highly acetylated on the K40 of α-tubulin [36,41] and polyglutamylated [42] and accumulated Δ2-tubulin [43]. The actions of spindle microtubules during mitosis is regulated by several MAPs [44] and assisted by several motor proteins [45]. For instance, the initial capture and transport of peripheral chromosomes by microtubules is mediated by dynein/dynactin [46,47,48,49], a minus-end-directed motor localized at unattached kinetochores [50,51], whereas the subsequent congression to the spindle equator is mediated by another kinetochore-associated motor, Centromere Protein E (CENP-E)/kinesin-7, with microtubule plus-end-directed activity [52,53]. Other mitotic motors include kinesin-5, which slides antiparallel microtubules to ensure proper centrosome separation, spindle bipolarity and spindle elongation during anaphase, as well as kinesin-13s, which lack motor activity but promote microtubule depolymerization to control spindle length and mediate mitotic error correction [54,55,56,57,58,59]. Thus, the mitotic spindle is an anisotropic and highly heterogeneous structure, with dynamic astral microtubules essentially tyrosinated, in contrast with more stable microtubule subpopulations, such as kinetochore and a fraction of interpolar microtubules, which accumulate detyrosinated, Δ2, acetylated and polyglutamylated tubulin. How these modifications impact the action of the different mitotic motors that assist chromosome segregation remains poorly understood.

### 2.1. A Navigation System Guides Chromosomes to the Spindle Equator

Although tubulin diversity in the mitotic spindle has been recognized for several decades, the respective functional relevance for mitosis remained unclear until recently. One crucial implication of the tubulin code hypothesis is the regulation of MAPs and motors by specific tubulin isotypes and PTMs [4]. Original work in neurons revealed that classic kinesin motors, such as Kinesin-1, are able to recognize and have a preference for microtubules with particular tubulin PTMs, namely detyrosination and acetylation [60,61]. Subsequently, α-tubulin detyrosination was shown to regulate mitotic chromosome congression to the metaphase plate by guiding the microtubule plus-end-directed motor CENP-E/kinesin-7 at kinetochores in human cells [36]. In contrast, the microtubule minus-end-directed motor dynein/dynactin that is also localized at unattached kinetochores [50,51] preferentially associates with tyrosinated microtubules [40,62,63,64], which favor the initiation of motion but are dispensable for subsequent dynein/dynactin processivity [63,64]. Thus, detyrosinated/tyrosinated α-tubulin regulates the activity of opposing kinetochore motors, establishing a navigation system for chromosomes that assists their congression to the spindle equator [65] (Figure 2). Accordingly, during the initial capture of chromosomes, dynein/dynactin counteracts the action of chromokinesins on chromosome arms to move peripheral chromosomes along tyrosinated astral microtubules towards the vicinity of the poles [66]. By transporting peripheral chromosomes to the poles where the microtubule destabilizing activity of Aurora A kinase is higher [67,68], dynein/dynactin prevents the formation of stable end-on kinetochore–microtubule attachments that would otherwise cause the random ejection of polar chromosomes by chromokinesins [65,66]. Once at the poles, Aurora A-mediated phosphorylation activates CENP-E at kinetochores of polar chromosomes [69], thus allowing their transport specifically along detyrosinated spindle microtubules towards the equator. In agreement, recent super-resolution coherent-hybrid stimulated emission depletion microscopy [70] of CENP-E-GFP revealed its exclusive association with stable kinetochore and interpolar microtubule bundles but not with tyrosinated astral microtubules [71]. Curiously, α-tubulin acetylation on K40, which is also enriched on stable spindle microtubules [41], does not interfere with polar chromosome congression [36]. While the potential contribution of other tubulin PTMs to chromosome congression remains unknown, these findings support a robust working model in which tyrosinated/detyrosinated microtubules guide peripheral chromosomes towards the spindle equator.

### 2.2. A Mitotic Error Code

The regulation of kinetochore microtubule dynamics is essential for error correction and the maintenance of genome stability, since it allows the establishment of amphitelic kinetochore-MT attachments that lead to chromosome biorientation relative to the spindle poles. Kinesin-13s, such as Kinesin superfamily 2b (Kif2b) and mitotic centromere-associated kinesin (MCAK), promote kinetochore microtubule dynamics, thus playing a key role in the correction of mal-oriented chromosomes with erroneous kinetochore-microtubule attachments (e.g., syntelic, in which both sister kinetochores are oriented towards a single spindle pole, and merotelic, where a single kinetochore is attached with microtubules oriented to both poles) and, ultimately, in the prevention of chromosome mis-segregation [55,72] (Figure 2). In agreement, stimulation of kinetochore microtubule dynamics in otherwise chromosomally unstable cancer cells by increasing kinesin-13 depolymerase activity reestablished chromosomal stability [55,73]. Building on the previous finding that MCAK’s microtubule depolymerizing activity is reduced four-fold in the presence of detyrosinated microtubules in vitro [74,75], it was recently shown that the mitotic error correction activity of MCAK and Kif2b is regulated by α-tubulin detyrosination [37]. Accordingly, the experimental depletion of TTL or overexpression of VASH1-SVBP, which caused a constitutive increase of α-tubulin detyrosination in the vicinity of the kinetochores, compromised error correction, leading to chromosome segregation errors. Importantly, α-tubulin detyrosination specifically impaired the MCAK-based error correction machinery located on centromeres/kinetochores, and it did so without affecting global kinetochore microtubule dynamics, suggesting that mitotic error correction is exquisitely sensitive to the detyrosinated state of α-tubulin that likely occurs at the individual microtubule level. These data support the existence of a “mitotic error code” in which α-tubulin detyrosination/tyrosination signals and regulates MCAK activity at centromeres/kinetochores to discriminate between correct and incorrect kinetochore-MT attachments during mitosis (Figure 2).

Complete centrosome separation before nuclear envelope breakdown prevents subsequent segregation errors and ensures mitotic fidelity [76]. This relies on several elements, including the microtubule motors kinesin-5, required for centrosome separation, and dynein/dynactin, which promote both centrosome separation and positioning [77,78]. Similar to dynein/dynactin, kinesin-5 appears to have increased affinity to tyrosinated dendritic microtubules in neurons [79], but direct evidence from in vitro reconstitution assays is still lacking. Nonetheless, recent work in which centrosome positioning in human mitotic cells was tracked in 3D indicated that centrosome separation at nuclear envelope breakdown is insensitive to the tyrosinated state of α-tubulin [37]. This reinforces the idea that the observed increase in mitotic errors associated with excessive α-tubulin detyrosination is due to the incapacity to correct, rather than an increased propensity to make errors.

### 2.3. Role in Mitotic Spindle Orientation and Positioning

Mitotic spindle orientation and positioning in the cell center is essential for accurate cell division and relies on the action of pulling forces on astral microtubules [80]. In particular, dynein/dynactin anchored to cortical proteins or cytoplasmic organelles was shown to play a significant role in spindle orientation/positioning [81,82,83], possibly through its increased affinity to tyrosinated astral microtubules (Figure 1). Indeed, modulation of the α-tubulin tyrosination state, either through TTL knockout [40] or CRISPR/Cas9-mediated editing of the C-terminal tyrosine [83], caused spindle orientation defects. In contrast, an experimental decrease of α-tubulin detyrosination after VASH1/2 silencing increased the depolymerase activity of MCAK, resulting in disoriented spindles, with shorter astral microtubules [19]. Taken together, these observations indicate that the mechanisms behind spindle orientation/positioning rely on the intrinsic nature (i.e., nonmodified) of tyrosinated α-tubulin to allow astral microtubules to establish a correct cell division plane (Figure 2).

### 2.4. Roles in Centrosome Structure and Cytokinesis

Tubulin polyglutamylation is highly enriched on centriole microtubules [42,84] and has been proposed to contribute to normal mitosis by maintaining centrosome structure [84,85]. Indeed, recent super-resolution imaging of the centriole structure revealed the specific distribution of polyglutamylation on centriole MTs and suggested a key role for this PTM in ultrastructural organization of specific centriolar proteins [86]. Furthermore, tubulin polyglutamylation promotes the activity of the microtubule-severing enzymes spastin and katanin [87,88,89,90], which are also implicated in cell division. Indeed, their activities regulate several cellular processes that likely impact chromosome segregation fidelity, such as microtubule poleward flux, spindle orientation and length [91,92,93]. Spastin and katanin are also required for the abscission step and completion of cytokinesis [94,95,96]. Like spastin [94] and katanin [96], polyglutamylated tubulin is enriched at the midbody [87], and a tubulin mutation that compromises polyglutamylation (and, possibly, also polyglycylation) in cilia was shown to cause cytokinesis defects [97]. These results suggest that the completion of cytokinesis relies on the regulation of spastin and katanin activities by tubulin polyglutamylation.

## 3. The Cancer Tubulin Code

### 3.1. (De)Regulation of Tubulin Isotypes and PTMs in Cancer 

Several works have reported an emerging link between alterations of tubulin isotypes and PTMs and/or associated modifying enzymes with certain cancers; most noticeable, those occurring in the breast, colon, prostate, liver, brain, bile duct and pancreas (Table 1). These alterations often correlate with specific cancer properties, including poor outcome/prognosis [98,99,100] and metastatic ability [98], supporting the potential use of cancer tubulin isotypes and/or PTM signatures as useful biomarkers, as well as for therapeutic purposes. However, a comprehensive and definitive view on the real potential is still lacking, especially concerning causality, since the available data is still limited and often contradictory.

### 3.2. Functional Implications of the Cancer Tubulin Code 

The differential regulation of specific tubulin isotypes and/or PTMs in cancer might reflect their role in key mechanisms underlying cell transformation (Figure 3). β3-tubulin (TUBB3) is the most frequent tubulin isotype associated with specific cancer features. Its expression was proposed to be important for tumor development [101,102] and metastatic ability [102,103], correlating with poor outcomes [103,104]. The expression of other isotypes such as β2-tubulin, and its altered cellular localization in colorectal cancer, also correlate with poor outcomes [105]. The differential expression of tubulin isotypes have been extensively associated with response to microtubule-targeting drugs, such as taxanes, commonly used in chemotherapy (reviewed in [106]). The origin of this link may be on the known regulation of microtubule dynamics by specific tubulin isotypes, with microtubules containing β3-tubulin being more dynamic compared to other β-tubulin isotypes [107,108,109].

In addition, the regulation of cell proliferation, which is essential for cancer development, was proposed to be mediated by certain tubulin PTMs. In this regard, the tubulin glycylase TTLL3 was proposed to restrict cell proliferation in the colon and is downregulated in colon cancer [110], whereas the tubulin glutamylase TTLL4 was suggested to promote cell proliferation in pancreatic cancer cells [111]. However, whether this was specifically due to a role of TTLL4 in tubulin glutamylation remains controversial, since an additional activity towards non-tubulin substrates has been reported [112,113]. The tubulin acetyltransferase αTat1 was also shown to be required for contact inhibition of cell proliferation in vitro [114]. In agreement, the tubulin deacetylase HDAC6 seems to promote cell proliferation in several cancer cell lines [115,116,117,118,119], consistent with its upregulation in some cancers (Table 1). Nevertheless, specificity remains to be demonstrated, since HDAC6 is also known to modulate the acetylation of other substrates besides tubulin [120]. Interestingly, the activity of TTLL3 and HDAC6 was also proposed to impact tumorigenesis. Accordingly, the experimental loss of TTLL3 in a mouse model of tumorigenesis resulted in the development of cancer, thus validating its downregulation in colon cancer and suggesting a cancer-suppressing role for tubulin glycylation [110]. In contrast, the expression of HDAC6 promoted colony and spheroid formation of cancer cells, as well as tumor growth in mice [115,116,118,119]. The activity of other tubulin-modifying enzymes, such as TTL, is also decreased during tumorigenesis in mouse models, resulting in increased detyrosinated- and Δ2-tubulin levels [121]. This is consistent with the association between α-tubulin detyrosination and tumor aggressiveness [99], as well as with the frequent downregulation of TTL and consequent upregulation of α-tubulin detyrosination in several cancers (Table 1).

The recent discovery of Vasohibins (VASH1 and VASH2) as TCPs [13,14] revitalized the discussion about the role of tubulin detyrosination in cancer. Vasohibins and their associated SVBP were originally identified as secreted proteins implicated in angiogenesis [127]. While VASH2 promotes vascularity by accumulating at the sprouting zone, VASH1 expression is increased in endothelial cells of the termination zone, where it inhibits vascularity [128]. During tumor development in mice xenograft models, experiments involving the administration of ectopic VASH1 indicated that it inhibits tumor lymphangiogenesis [129], angiogenesis and growth [130]. On the other hand, VASH2, which appears to play an important role in cancer cell proliferation [124], promotes tumor angiogenesis and growth [124,131,132,133]. Noteworthy, none of these studies demonstrate that the observed impact in cancer was due to defective tubulin detyrosination. However, human patients suffering from a broad range of carcinomas had mutations in VASH1 and VASH2 that compromised their tubulin detyrosination activity [17] and, more recently, it was suggested that the MT detyrosinating activity of VASH1 inhibited angiogenesis by interfering with endocytosis and trafficking of proangiogenic factor receptors [134]. Taken together, these findings suggest that, in addition to the downregulation of TTL [121], the link between tubulin detyrosination and tumorigenesis may be attributed to the role of Vasohibins in angiogenesis. The availability of VASH1/2-SVBP knockout mice [128,135] will be instrumental in clarifying the apparently opposite roles of VASH1 and VASH2 in cancer and whether this is due to their secreted and/or tubulin detyrosinating activities.

### 3.3. The Cancer Tubulin Code in Cell Migration and Invasion 

Tubulin PTMs have also been implicated in epithelial-to-mesenchymal transition (EMT), a key process behind metastasis initiation. For instance, experimental increase of the tubulin deacetylase HDAC6 promoted EMT, whereas TGF-β induction of EMT downregulated tubulin acetylation [136]. Likewise, the induction of EMT also correlated with the downregulation of TTL and the consequent increase of tubulin detyrosination [123], as shown before during tumor development [121], thus pointing to the possible involvement of these tubulin PTMs and associated enzymes in cell transformation.

Interestingly, tubulin acetylation is also frequently associated with the regulation of cell migration, although this remains controversial. While HDAC6 expression and activity was proposed to promote cell migration [12,120,136,137,138], the opposite effect was observed after the loss of αTat1 or mutation of the α-tubulin lysine 40 (K40R) [98,137,139,140]. The establishment of cell adhesion to the substrate also has implications for cell motility, and the loss of either HDAC6 or αTat1 leads to an increased focal adhesion area and number, respectively, as well as decreased dynamics [137,141]. However, other works reported that a loss of αTat1 leads to a decrease in the focal adhesion number [114]. The basis for this discrepancy remains unclear, but it is likely associated with different experimental setups; one study investigated the role of αTat1 in wound-induced migrating cells [137], while the other used normally growing cells [114], raising the possibility that αTat1 promotes focal adhesion dynamics specifically during cell migration. 

The upregulation of tubulin acetylation in metastatic breast tumors and cell lines [98] is consistent with its association with cancer cell invasiveness. RNAi-mediated depletion of either αTat1 or HDAC6 indicated that their expression induced breast cancer cell invasion [139,140,142]. Additionally, the increased tubulin acetylation of these metastatic breast cancer cells promoted micro-tentacle generation and cell reattachment ability, essential for metastasis [98]. Likewise, a high frequency of micro-tentacles and cell reattachment were also associated with tubulin detyrosination [123,143]. Collectively, these data favor a potential role of tubulin acetylation in metastasis progression. While HDAC6 indiscriminately acts upon multiple protein targets, the direct modulation of tubulin acetylation by K40R mutation experiments suggest that the upregulation of tubulin acetylation is a metastasis-promoting factor, supporting the αTat1-related findings. This would explain the link between the upregulation of tubulin acetylation and poor prognosis in breast cancer patients [98], but unspecific effects due to the overexpression of GFP-tagged K40R mutant α-tubulin cannot be excluded. 

## 4. How Alterations of the Tubulin Code in Mitosis Might Be Implicated in Cancer

Chromosomal instability, a hallmark of cancers, has been shown to promote the metastatic process [73]. Indeed, the overexpression of Kif2b or MCAK, in addition to reestablishing the stability of chromosomally unstable cancer cells [55,73], inhibits metastasis in vitro and in vivo, with a consequent increase in survival [73]. Given that excessive tubulin detyrosination might lead to chromosomal instability by suppressing the error correction activity of MCAK and Kif2b [37], together with the observed upregulation of tubulin detyrosination in invasive cancer and with poor prognosis (Table 1), it raises the exciting possibility that an increase in tubulin detyrosination might promote cancer progression through inhibition of the mitotic error correction machinery. However, an extensive analysis of tubulin detyrosination in chromosomal instability-prone cancers, together with the elucidation of its implications for cancer metastasis, is necessary for its establishment as potential diagnostic and prognostic biomarkers. In addition, tubulin detyrosination represents a promising therapeutic target for cancer suppression—for example, by using TCP inhibitors, such as epoY [13] or parthenolide [144]. 

The deregulation of tubulin detyrosination in cancers might also be involved in other mitotic-related cancer features. Firstly, the cell cycle delay observed upon VASH1/2 [19] and VASH2 [124] deletion might unveil the importance of VASH2 for proper cancer cell proliferation and tumor development [124,131,132,133]. Furthermore, both experimental upregulation and downregulation of tubulin detyrosination led to congression defects, causing alterations in the CENP-E-mediated transport of chromosomes to the spindle equator [36]. Additionally, the decrease of CENP-E expression is well-established to promote mild chromosomal instability and aneuploidy, as well as tumorigenesis in mice [145,146,147,148]. Therefore, the deregulation of tubulin detyrosination in cancers (Table 1) may also account for cancer promotion under conditions of moderate chromosomal instability, such as those associated with mild problems in chromosome congression. Further investigation is required to fully understand the potential implications of tubulin detyrosination and other PTMs for tumorigenesis and the respective link with chromosomal instability.

## 5. Conclusions and Outlook

Since the initial observations implicating tubulin PTMs in cell division, recent works have allowed a deeper understanding of their involvement—in particular, detyrosination/tyrosination—in the coordination of several mechanisms underlying faithful chromosome segregation during mitosis. However, considerable knowledge is still lacking in order to establish a complete picture of the roles played by the tubulin code in mitosis. The scenario is no different regarding the emerging cancer tubulin code, in which a considerable amount of disconnected data dominates. Nevertheless, there are already some promising links between the deregulation of certain tubulin isotypes and PTMs (notoriously, acetylation, detyrosination and glycylation) and several cancers. A more systematic investigation of these links will be of high priority to the field and might prove important for diagnostic and prognostic purposes. This is likely to have a major impact in understanding and mitigating the acquired resistance to microtubule-targeting drugs, the biggest threat in current cancer chemotherapy. Future work is also necessary to establish clear functional links beyond correlations by taking advantage of emerging molecular tools and model systems for the modulation and analysis of tubulin isotypes and PTMs (both in vitro and in vivo) that will strengthen and clarify their potential therapeutic value for the treatment of human cancers.

## Figures and Tables

**Figure 1 cells-09-02356-f001:**
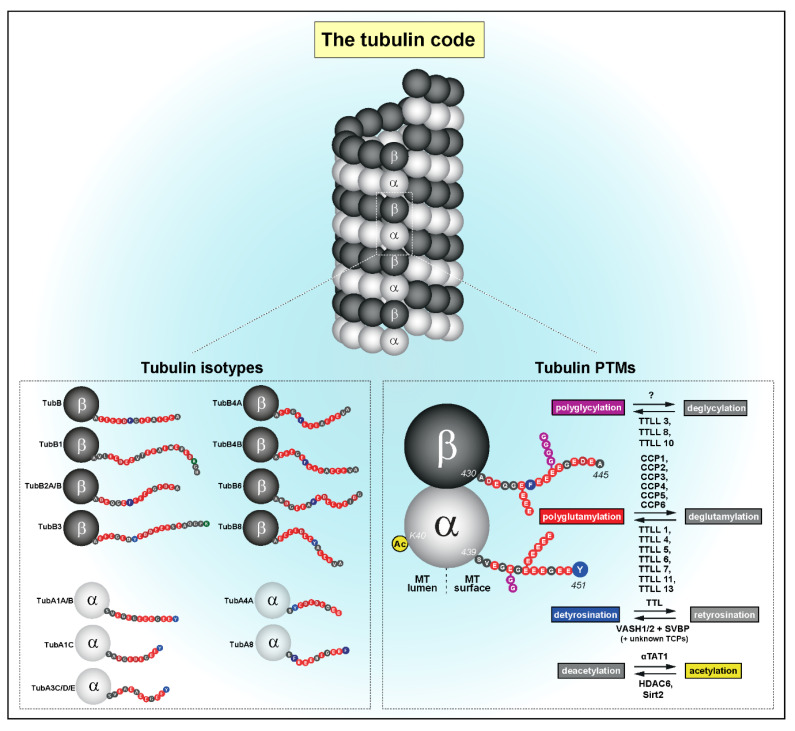
The tubulin code combines different tubulin isotypes and post-translational modifications (PTMs) to generate microtubule diversity. Only the best-characterized isotypes and PTMs (+ respective enzymes) are depicted. See main text for details.

**Figure 2 cells-09-02356-f002:**
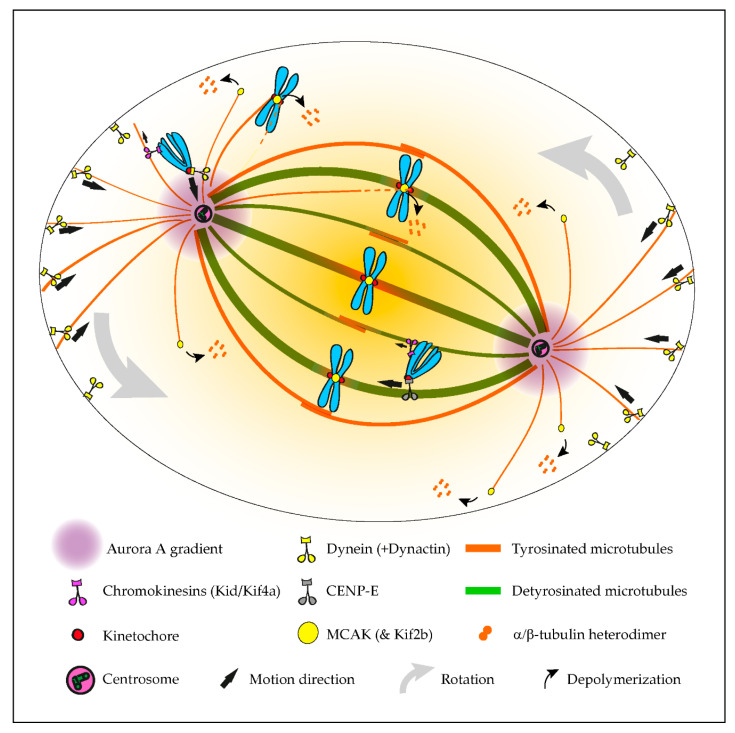
Summary of the established roles of the tubulin code in mitosis. The initial capture of peripheral chromosomes is mediated by dynein/dynactin at kinetochores, upon which, the chromosome is brought to the vicinity of the centrosome by lateral transport along tyrosinated astral microtubules. This prevents the random ejection of the chromosome by the action of Chromokinesins on chromosome arms. Once at the pole, high Aurora A activity prevents the stabilization of end-on kinetochore-microtubule attachments, which otherwise would favor the action of Chromokinesins on chromosome arms. In parallel, Aurora A-mediated phosphorylation activates CENP-E at kinetochores. This initiates transport towards the spindle equator (congression) along stable detyrosinated microtubules. Mitotic centromere-associated kinesin (MCAK) and Kif2b (not depicted) at centromeres and kinetochores are also inhibited by tubulin detyrosination on kinetochore microtubules, allowing the correction of syntelic and merotelic attachments, while preserving correct amphitelic attachments on bi-oriented chromosomes. MCAK at the microtubule plus ends also regulates astral microtubule length to allow interaction with dynein/dynactin at the cortex or cytoplasmic organelles (not depicted), which exerts pulling forces necessary for spindle orientation and positioning. See main text for details.

**Figure 3 cells-09-02356-f003:**
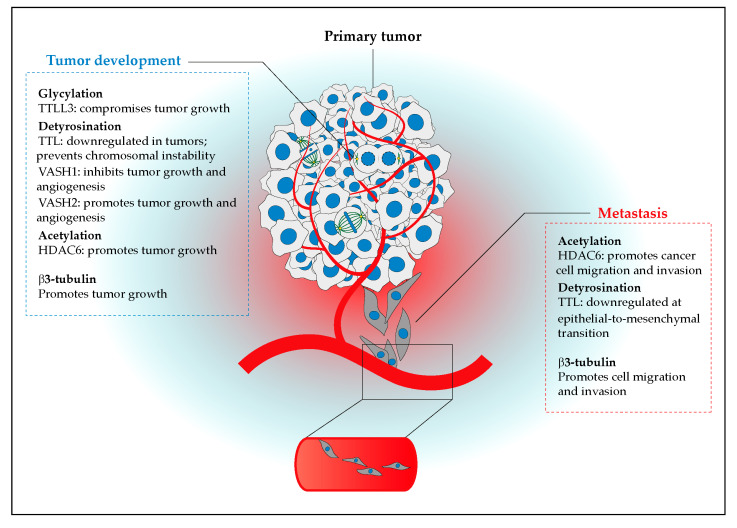
Implications of the tubulin code for tumor progression and metastasis. While the downregulation of TTLL3 (glycylation), together with the expression of VASH2 (detyrosination), HDAC6 (acetylation) and β3-tubulin, promotes tumor growth, this is inhibited by VASH1 (detyrosination). Tumor formation and chromosomal instability is also associated with the downregulation of TTL. Tubulin acetylation, detyrosination and β3-tubulin isotypes might promote several steps of metastasis associated with the epithelial-to-mesenchymal transition, such as cell migration and invasion. See main text for details.

**Table 1 cells-09-02356-t001:** Tubulin isotypes, post-translational modifications and modifying enzymes in cancer.

Tubulin PTM (and/or Enzymes)/Isotype	Cancer	Regulation	References
Detyrosination	Prostate Cancer Cells	Upregulated	[122]
	Poor Prognosis Breast Tumors	Upregulated	[99]
	Invasive Ductal Carcinoma (Breast)	Upregulated	[123]
TTL	Prostate Cancer Cells	Downregulated	[122]
	Poor Prognosis Neuroblastomas	Downregulated	[100]
VASH2	Hepatocellular carcinoma Tissues and Cell Lines	Upregulated	[124]
Δ2-Tubulin	Prostate Cancer Cells	Downregulated	[122]
Acetylation	Metastatic Breast Tumors and Cell Lines	Upregulated	[98]
HDAC6	Pancreatic Tumors	Upregulated	[125]
	Glioblastoma Tissues and Cell Lines	Upregulated	[118]
	Cholangiocarcinoma Cell Lines	Upregulated	[115]
Glutamylation/ Polyglutamylation	Prostate Cancer Cells	Upregulated	[122]
TTLL4	Pancreatic Ductal Adenocarcinoma Cells	Upregulated	[111]
Glycylation TTLL3	Colon Tumors and Cell Lines	Downregulated	[110]
β3-tubulin	Pancreatic Tumors and Cell Lines	Upregulated	[102]
	Pancreatic Ductal Adenocarcinoma Tissues	Upregulated	[126]
	Breast Cancer Brain Metastases	Upregulated	[103]

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
