# Peer review of "The Tubulin Code in Mitosis and Cancer"

_cells, 2020, doi:10.3390/cells9112356_

Round 1
Reviewer 1 Report
Lopes and Maiato have written a very insightful review in which they connect the many lose ends that hint towards a key role of the tubulin code in cancer. This is a very important issue, and not at all broad knowledge, which is why this review is timely and novel.
The review is crisp, easy to read, and mostly logically structured. The illustrations are of good quality and give a nice overview of the concepts discussed.
Overall, while the review is written in a very good English, however, some re-reading would help to spot instances at which the sentences are not entirely self-explanatory, e.g. using too much of specialist language assuming the reader is familiar with the tubulin code. As one of the key audiences of this review should be the community of cancer researchers, which are mostly unfamiliar with this terminology, it is very important to make sure concepts are (even repetitively) explained in a clear manner.
Specific points:
- Throughout the review, the authors try to coin the terms “cancer tubulin code” and “mitotic tubulin code”. This insinuates that there are specific, clearly defined “codes” on tubulin in cancer and in mitosis respectively. However, this is not the case. There is one concept of a “tubulin code”, which can then have certain implications in cancer or mitosis. It is therefore strongly recommended to not use these terms, but so talk rather about tubulin code in cancer / in mitosis.
- Following point (1), the abstract line 21/22 should read “…mitosis, together with its emerging roles in cancer….”
- The introduction gives a nice overview of the tubulin code. It might be helpful for somebody reading for the first time of this concept to have a figure illustrating the different elements of the tubulin code.
- Lies 54/55: this sentence is hard to understand. The authors certainly wanted to say something along these lines: “…occur at the tubulin core structure as opposed to the CTTs”
- The notion of “tubulin tyrosination” is known to generate misunderstandings, as tyrosination is NOT a PTM given that most alpha-tubulin genes carry a gene-encoded Tyr. Obviously, once tubulin becomes detyrosinated, or TUBA4A is expressed, tyrosination can be considered a PTM. To avoid confusing the reader, it will be important to clearly state that the ground-state of alpha-tubulin is tyrosinated. In line 61, where the authors write “…mitotic spindle microtubules are vastly tyrosinated.”, they might want to specify that this means non-modified (by detyrosination).
- Line 109: “Building on the previous recognition that MCAK´s microtubule…” reads a bit strange, what about “Based on the previous finding that MCAK´s microtubule…”?
- Lines 125-126: the authors propose that both, dynein/dynactin and kinesin-5 have increased affinity to tyrosinated microtubules. While for dynein/dynactin this has been firmly established with reconstitution experiments, it might be important to discuss, or mention, how solid the evidence for kinesin-5 is. Is it possible that these data are obtained by rather indirect measures?
- Line 160: this chapter vaguely talks about centrosome structure and cytokinesis, but what it is really about is the current knowledge on what tubulin glutamylation does during cell division. In the centrosome part, the authors might want to mention recent studies by the Guichard team showing the specific distribution of polyglutamylation on centrioles. Their work suggests a key function in arranging the centriolar proteins, thus arranging the ultrastructure of the centrioles (and thus, affecting their functions).
(Mahecic D, Gambarotto D, Douglass KM, Fortun D, Banterle N, Ibrahim KA, Le Guennec M, Gonczy P, Hamel V, Guichard P, Manley S (2020) Homogeneous multifocal excitation for high-throughput super-resolution imaging. Nat Methods 17: 726-733) - Lines 183-184: this sentence sounds like a circular argument. Did the authors wanted to say this: “The differential regulation of specific tubulin PTMs in cancer might reflect their role in key mechanisms underlying cancer.”?
- Line 189: the fact that TTLL4 modifies tubulin is not controversial, it rather needs to be verified each time whether the effects of TTLL4 are related to the modification of tubulin or other substrates.
- Line 196: “…decreased, resulting in increased detyrosinated- and Δ2-tubulin levels.”
- Lines 195-199 should be moved down after current line 204. It is not intuitive to follow the text when it constantly switches between tubulin PTMs. Especially for readers who are not familiar with the field and the (rather confusing) names of the enzymes.
- In line 205, the authors mention vasohibins, in line 207 VASH2. Despite the fact that these enzyme names were defined before in the review, it might be helpful to repeat that VASH2 is a vasohibin.
- Line 209: what are “gene modulation experiments”? The authors should be more precise and describe briefly what the experiments are.
- Lines 225-232: The authors talk about discrepancies between studies. Is there a way to speculate what could cause them? Perhaps by mentioning experimental approaches used in the different studies and how they could lead to misinterpretation?
- Along these lines, the authors mention in line 241 the experiments with alpha-tubulin mutated at the K40 residue to avoid acetylation. This is a clean experiment in one sense, but there are a couple of problems related with this: a) most of the times the protein is overexpressed in cells, thus being present in addition to normal, acetylatable tubulin, b) in a number of studies, the mutant alpha-tubulin is expressed as N-terminal fusion with GFP. However, it is known that GFP-tubulin incorporates only partially into the microtubule lattice. This should be taken into consideration when discussing these experiments, and clearly stated (if it is the case) as their limitation.
- Conclusions: the authors conclude correctly that recent advances have helped to understand the role of detyrosination/tyrosination in cell cycle control. However, they only marginally mention in the conclusion the mass of unconnected data on the role of different PTMs in cancer. As this was one key part of this review, they should make sure the conclusion is balanced, and reflects the content of the review. Perhaps by mentioning which PTMs were connected with cancer, and what this should guide us to?
- Figure 1: The way it is shown suggests that kinetochore microtubules are fully detyrosinated. However, in reality they also contain tyr-tubulin, so they are only partially detyrosinated. Is there a way to show this without making the figure too complex?
- Figure 2 and legend: TTLL3: “prevention of tumour growth” is an overstatement. Perhaps “affects tumour growth” or “affects proliferation” could be used instead.
Author Response
- Throughout the review, the authors try to coin the terms “cancer tubulin code” and “mitotic tubulin code”. This insinuates that there are specific, clearly defined “codes” on tubulin in cancer and in mitosis respectively. However, this is not the case. There is one concept of a “tubulin code”, which can then have certain implications in cancer or mitosis. It is therefore strongly recommended to not use these terms, but so talk rather about tubulin code in cancer / in mitosis.
R: We have re-written the abstract and manuscript text to avoid confusion with terminology and removed the term “mitotic tubulin code”. We nevertheless consider that the notion of a “cancer tubulin code” is an easy grasping concept that will appeal to medical researchers in the cancer field and raise awareness for the topic.
- Following point (1), the abstract line 21/22 should read “…mitosis, together with its emerging roles in cancer….”
R: This has been re-written for clarity
- The introduction gives a nice overview of the tubulin code. It might be helpful for somebody reading for the first time of this concept to have a figure illustrating the different elements of the tubulin code.
R: A new figure summarizing the tubulin code is now provided
- Lies 54/55: this sentence is hard to understand. The authors certainly wanted to say something along these lines: “…occur at the tubulin core structure as opposed to the CTTs”
R: This has been re-written for clarity
- The notion of “tubulin tyrosination” is known to generate misunderstandings, as tyrosination is NOT a PTM given that most alpha-tubulin genes carry a gene-encoded Tyr. Obviously, once tubulin becomes detyrosinated, or TUBA4A is expressed, tyrosination can be considered a PTM. To avoid confusing the reader, it will be important to clearly state that the ground-state of alpha-tubulin is tyrosinated. In line 61, where the authors write “…mitotic spindle microtubules are vastly tyrosinated.”, they might want to specify that this means non-modified (by detyrosination).
R: This has been re-written for clarity
- Line 109: “Building on the previous recognition that MCAK´s microtubule…” reads a bit strange, what about “Based on the previous finding that MCAK´s microtubule…”?
R: This has been re-written for clarity
- Lines 125-126: the authors propose that both, dynein/dynactin and kinesin-5 have increased affinity to tyrosinated microtubules. While for dynein/dynactin this has been firmly established with reconstitution experiments, it might be important to discuss, or mention, how solid the evidence for kinesin-5 is. Is it possible that these data are obtained by rather indirect measures?
R: We agree that the evidence for kinesin-5 is indirect and have re-written this sentence for clarity
- Line 160: this chapter vaguely talks about centrosome structure and cytokinesis, but what it is really about is the current knowledge on what tubulin glutamylation does during cell division. In the centrosome part, the authors might want to mention recent studies by the Guichard team showing the specific distribution of polyglutamylation on centrioles. Their work suggests a key function in arranging the centriolar proteins, thus arranging the ultrastructure of the centrioles (and thus, affecting their functions).
(Mahecic D, Gambarotto D, Douglass KM, Fortun D, Banterle N, Ibrahim KA, Le Guennec M, Gonczy P, Hamel V, Guichard P, Manley S (2020) Homogeneous multifocal excitation for high-throughput super-resolution imaging. Nat Methods 17: 726-733)
R: We thank the reivewer for pointing out this study, which escaped our attention. The study is now cited and discussed.
- Lines 183-184: this sentence sounds like a circular argument. Did the authors wanted to say this: “The differential regulation of specific tubulin PTMs in cancer might reflect their role in key mechanisms underlying cancer.”?
R: This has been re-written for clarity
- Line 189: the fact that TTLL4 modifies tubulin is not controversial, it rather needs to be verified each time whether the effects of TTLL4 are related to the modification of tubulin or other substrates.
R: This has been re-written for clarity
- Line 196: “…decreased, resulting in increased detyrosinated- and Δ2-tubulin levels.”
R: This has been corrected
- Lines 195-199 should be moved down after current line 204. It is not intuitive to follow the text when it constantly switches between tubulin PTMs. Especially for readers who are not familiar with the field and the (rather confusing) names of the enzymes.
R: This has been corrected as suggested.
- In line 205, the authors mention vasohibins, in line 207 VASH2. Despite the fact that these enzyme names were defined before in the review, it might be helpful to repeat that VASH2 is a vasohibin.
R: This has been re-written for clarity
- Line 209: what are “gene modulation experiments”? The authors should be more precise and describe briefly what the experiments are.
R: This is now clarified
- Lines 225-232: The authors talk about discrepancies between studies. Is there a way to speculate what could cause them? Perhaps by mentioning experimental approaches used in the different studies and how they could lead to misinterpretation?
R: We now point out some possible reasons.
- Along these lines, the authors mention in line 241 the experiments with alpha-tubulin mutated at the K40 residue to avoid acetylation. This is a clean experiment in one sense, but there are a couple of problems related with this: a) most of the times the protein is overexpressed in cells, thus being present in addition to normal, acetylatable tubulin, b) in a number of studies, the mutant alpha-tubulin is expressed as N-terminal fusion with GFP. However, it is known that GFP-tubulin incorporates only partially into the microtubule lattice. This should be taken into consideration when discussing these experiments, and clearly stated (if it is the case) as their limitation.
R: We now discuss the possibilities suggested by the reviewer.
- Conclusions: the authors conclude correctly that recent advances have helped to understand the role of detyrosination/tyrosination in cell cycle control. However, they only marginally mention in the conclusion the mass of unconnected data on the role of different PTMs in cancer. As this was one key part of this review, they should make sure the conclusion is balanced, and reflects the content of the review. Perhaps by mentioning which PTMs were connected with cancer, and what this should guide us to?
- Figure 1: The way it is shown suggests that kinetochore microtubules are fully detyrosinated. However, in reality they also contain tyr-tubulin, so they are only partially detyrosinated. Is there a way to show this without making the figure too complex?
R: We tried to improve our figure as suggested.
- Figure 2 and legend: TTLL3: “prevention of tumour growth” is an overstatement. Perhaps “affects tumour growth” or “affects proliferation” could be used instead.
R: This has been corrected.
Reviewer 2 Report
The review manuscript by Lopes and Maiato summarizes the present knowledge on tubulin post-translational modifications (PTMs) with special focus on mitosis and PTMs-linked defects of mitosis involved in cancer. This is a short informative review providing a comprehensive overview about the roles of modified microtubules (more specifically detyrosinated/tyrosinated) in the different processes occurring during mitosis, and for tumor development and invasion.
I have a few minor concerns/comments to enhance the clarity of the manuscript:
- Line 66-69 and figure 1. There are dynamic interpolar Tyr-microtubules (which are probably different from the deTyr/delta2 microtubules, and a majority), as shown in previous studies (Gundersen/Bulinski old articles, Peris et al J Cell Biol 2006). This should be specified in text and added in figure (some red with the green). The current presentation generates the incorrect feeling that interpolar microtubules are all modified.
- In beginning of part 2, a short introduction on the main molecular actors of mitosis which are described later for their link to specific PTMs (CENP-E/kinesin-7, Kinesins-13s, etc..), together with their specific role(s) in mitosis (and reference to figure 1) would help.
- Figure 1. The review would benefit from a clarification of this figure. Kinesin/dynein motors are too small, colors cannot be clearly seen, and thus the different motors cannot be distinguished. Please magnify them. Annotating in figure and legend the different processes described would help: (a) initial capture, (b) at the pole, etc… Also, the yellow circle is confusing. It represents kinetochores and centrosomes, and the fact that MCAK and Kif2b are enriched within them. Please clarify.
- Paragraph lines 205-216. Please discuss the proposed role of VASHs as a secreted factor, which may act from outside the cells to promote cancer. Then, the link to detyrosination in cancer (with what is currently published) is not obvious. To note however, human patients suffering from different types of carcinomas show mutations in VASH1 and VASH2 leading to the alteration of detyrosination activity were described in the recent work by Wang et al (Nature Struc. Mol. Biol. 2019). Also, to note, the only described modifying enzyme that is, with no doubt, tubulin-specific is TTL (HDAC6, by instance, has other targets than tubulin, …). All this would be worthy of explanation, and speculation.
- As this review may be read by non-experts or students, use of abbreviations should be limited. Please, use only “PTMs”, and the names of specific proteins (such as MCAK, CENP-E). Other abbreviations, such as CTT, kMT, NEBD, CIN, are not necessary and should be eliminated.
- Line 47. The fact that CCPs can also lead to the formation of ∆3-tubulin in cells published by Aillaud et al MBoC 2016) might be mentioned.
- Line 52. It should be mentioned that deglycylases are still unknown, and that not all detyrosinases are presently identified.
- It might be better to always use detyrosination/tyrosination, and not alternatively detyrosination/tyrosination and tyrosination/detyrosination (Lines 98 &142)
- Please correct or define the following terms. Line 30, aka? Lines 78, 92, 103, please define CENP-E, CH-STED, MCAK.
- Figure 2 and lines-237-239. Why not add detyrosination in the figure under metastasis, which is described as involved in text?
- Line 54-55. ‘PTMs such as … occur in the core structure outside the CTTs of α- and β-tubulins. ‘Adjacent’ would be better than ‘outside’.
- The following sentences are not understandable. Please, clarify. Line 186: ‘TTLL3 was proposed to contain cell proliferation…’. Contain? Line 240: ‘HDAC6 activity is promiscuous’. Promiscuous?
- Line 239-240. Collectively, these data reinforce the potential of tubulin acetylation in metastasis progression. ‘Argue/favor a potential role of tubulin acetylation…’ would be more understandable.
Author Response
- Line 66-69 and figure 1. There are dynamic interpolar Tyr-microtubules (which are probably different from the deTyr/delta2 microtubules, and a majority), as shown in previous studies (Gundersen/Bulinski old articles, Peris et al J Cell Biol 2006). This should be specified in text and added in figure (some red with the green). The current presentation generates the incorrect feeling that interpolar microtubules are all modified.
R: The figure was modified to reflect the suggestions of the reviewer.
- In beginning of part 2, a short introduction on the main molecular actors of mitosis which are described later for their link to specific PTMs (CENP-E/kinesin-7, Kinesins-13s, etc..), together with their specific role(s) in mitosis (and reference to figure 1) would help.
R: Proper introduction to the main motors implicated in mitosis is now provided.
- Figure 1. The review would benefit from a clarification of this figure. Kinesin/dynein motors are too small, colors cannot be clearly seen, and thus the different motors cannot be distinguished. Please magnify them. Annotating in figure and legend the different processes described would help: (a) initial capture, (b) at the pole, etc… Also, the yellow circle is confusing. It represents kinetochores and centrosomes, and the fact that MCAK and Kif2b are enriched within them. Please clarify.
R: The requested changes have been implemented in figure 1.
- Paragraph lines 205-216. Please discuss the proposed role of VASHs as a secreted factor, which may act from outside the cells to promote cancer. Then, the link to detyrosination in cancer (with what is currently published) is not obvious. To note however, human patients suffering from different types of carcinomas show mutations in VASH1 and VASH2 leading to the alteration of detyrosination activity were described in the recent work by Wang et al (Nature Struc. Mol. Biol. 2019). Also, to note, the only described modifying enzyme that is, with no doubt, tubulin-specific is TTL (HDAC6, by instance, has other targets than tubulin, …). All this would be worthy of explanation, and speculation.
R: This is an excellent point and we thank the reviewer for pointing it out. We have re-written the text to reflect the reviewer’s suggestions.
- As this review may be read by non-experts or students, use of abbreviations should be limited. Please, use only “PTMs”, and the names of specific proteins (such as MCAK, CENP-E). Other abbreviations, such as CTT, kMT, NEBD, CIN, are not necessary and should be eliminated.
R: The indicated abbreviations were eliminated.
- Line 47. The fact that CCPs can also lead to the formation of ∆3-tubulin in cells published by Aillaud et al MBoC 2016) might be mentioned.
R: This reference and mention to ∆3-tubulin is now provided.
- Line 52. It should be mentioned that deglycylases are still unknown, and that not all detyrosinases are presently identified.
R: This is now mentioned in the text.
- It might be better to always use detyrosination/tyrosination, and not alternatively detyrosination/tyrosination and tyrosination/detyrosination (Lines 98 &142)
R: This is now corrected as suggested.
- Please correct or define the following terms. Line 30, aka? Lines 78, 92, 103, please define CENP-E, CH-STED, MCAK.
R: These have now been corrected and defined.
- Figure 2 and lines-237-239. Why not add detyrosination in the figure under metastasis, which is described as involved in text?
R: This is now added.
- Line 54-55. ‘PTMs such as … occur in the core structure outside the CTTs of α- and β-tubulins. ‘Adjacent’ would be better than ‘outside’.
R: This has been corrected as suggested.
- The following sentences are not understandable. Please, clarify. Line 186: ‘TTLL3 was proposed to contain cell proliferation…’. Contain? Line 240: ‘HDAC6 activity is promiscuous’. Promiscuous?
R: This has now been clarified/corrected in the text.
- Line 239-240. Collectively, these data reinforce the potential of tubulin acetylation in metastasis progression. ‘Argue/favor a potential role of tubulin acetylation…’ would be more understandable.
R: This is now corrected as suggested.
Reviewer 3 Report
This is a very comprehensible review of our current knowledge on tubulin modifications, and on their cellular role. The manuscript describes the different forms of modifications, and discusses the impact of tubulin modifications on spindle microtubule dynamics, on spindle orientation, on chromosome segregation, on the correction of mitotic errors, and on the viability and invasive properties of cancer cells. Overall, the content of the review is up-to-date, complete, and well balanced. The text is well written and should be understandable to a wide readership. I have no particular criticism.
Author Response
This reviewer had no particular criticism.
Reviewer 4 Report
Submitted paper of Lopes and Maiato provides an overview on tubulin post-translational modifications and on the impact to mitotic processes with specific focus to chromosome instability. Manuscript contributes to elucidation of cancer tubulin signatures and deepens our knowledge of a link between tubulin PTMs, MAPs modifications and certain cancers. Understanding mutations of tubulin isotypes expressed specifically in tumors and their post-translational modifications might help to identify precise molecular targets for design of novel anti-microtubular drugs.
Authors provide a comprehensive overview on cancer tubulin code in mitosis and in cell migration and invasion. An expression of specific tubulin isotypes of βII tubulin and βIII tubulin and their nuclear localization as a prognostic marker of metastatic tumors should be mentioned.
I found the manuscript particularly timely and the paper is concise and well written.
Author Response
An expression of specific tubulin isotypes of βII tubulin and βIII tubulin and their nuclear localization as a prognostic marker of metastatic tumors should be mentioned.
R: This is now mentioned and readers directed to a recent review on tubulin isotypes in cancer.
Reviewer 5 Report
Summary
In the review “The tubulin code in mitosis and cancer” Danilo Lopes and Helder Maiato provide an updated view of the current knowledge on the regulation of mitotic functions by the “tubulin code” and discuss about the implications of these regulations in the pathological context of cancers.
The review first introduce the tubulin code and then make a specific focus on most relevant posttranslational modifications (PTMs) involved in mitosis. Then, review provide a recent survey of evidences for the implication of some PTMs (especially detyrosination) in controlling key regulators of chromosome alignment, segregation and mitotic spindle positioning.
The authors next focus on the regulation of mitotic functions by PTMs in the specific context of tumorigenesis and gather the current knowledge on identified correlation between tubulin PTMs and cancers. In that line, they also provide an interesting and objective discussion on the current understanding, controversial interpretation found in the literature and challenge to overcome.
Strengths:
The review is well written, documented, appropriately illustrated and accessible. Posttranslational modifications of tubulin, and more generally the so-called “tubulin code”, are becoming unavoidable topics in the microtubule cytoskeleton field to which the authors’ group has brought significant contributions.
Several reviews have already documented the link between the tubulin code and pathologies in general with a broader point-of-view but this review provides a more specific (and to my knowledge unique) focus on chromosome instability and cancer. The review offers a good balance of the different or controversial interpretations found on the literature on the role of PTMs in regulating mitosis or with their potential link with chromosome instability and cancer.
Conclusion:
The manuscript is suitable for publication. The minor points listed below will modestly help the manuscript to be more accurate or more accessible to a broad audience.
Minor comments:
- The authors stated in the abstract that the tubulin code is a combination of PTMs and tubulins isoforms. I totally agree with this definition and I think that it represents the most commonly accepted and accurate definition to date. However, Table 1 and more generally the review are focused on tubulin PTMs only. There are several examples of links between tubulin isoforms expression (e.g. beta-III) and cancer prognosis or drug resistances (for example reviewed here doi: 3390/ijms18071434). There are also experimental evidences that tubulin isoforms affect microtubule dynamic properties. The review is still relevant and there is already a lot to mention only on PTMs but I think this point must be clarified either by mentioning the knowledge on tubulin isoforms (which might be too long and out of the scope of this review) or by assuming the focus on (some) PTMs and therefore only on a truncated version of the code. In that case, the words “the tubulin code” and especially “the cancer tubulin code” although very trendy might be misleading and overstated. As the authors said on the conclusion, a lots remains to be done to understand the “complete mitotic tubulin code” but I’m afraid that reducing the tubulin code to PTMs would be inaccurate. The cancer tubulin code (if there is one) can certainly not be reduced to PTMs. Maybe the use of “A code” instead of a “The code…” or any other way to clarify this statement will be fair and useful to not compromise the future use and the validity of the term “tubulin code” and thence, the value of this review.
- Line #60-61 the authors make the link between dynamic microtubules and tyrosination. Since the review might be addressed to readers not necessarily familiar with tubulin posttranslational modifications of tubulin and all the different reactions involved, it might be useful to explain slightly more this cycle. Reactions specificity on soluble versus polymeric tubulin and the link between dynamic microtubules and tyrosinated state might not be obvious for everyone.
- Line #66, I see what the authors meant here with the difference between astral and spindle microtubules but the word “asymmetric” to describe the spindle could be confusing here since the two “half” spindles in the bipolar structure are symmetric (like in Figure 1).
- Line #169, the mutation that compromise cytokinesis have been shown in the ciliate Tetrahymena which represents a particular context in terms of microtubule populations and post-translational modifications compare to other systems like mammalian cells. Although the interpretation is correct, the authors might want to state here that this phenotype was observed in a ciliate especially since the mutation in this context can also affect (poly)glycylation.
- Line #190 when mentioning non-tubulin substrates of polyglutamylation and of TTLL4 enzyme, the reference [85] is relevant for cancer but the modification of non-tubulin substrates by TTLL4 was to my knowledge first reported in 2008 by van Dijk et al. 2008 JBC “Polyglutamylation Is a Post-translational Modification with a Broad Range of Substrates”. The modification of other substrates such as nucleosome assembly proteins (NAPs) was reported in 2000 by Regnard et al. 275, 15969 –15976
Comments on Figures
Figures and tables a simple and clear enough. However, given the quite large size of Figure 1, I think that an effort can be made to better visualize some details such as CENP-E motors that are really small. The legend for the orange dots (kinetochore) is missing and the different arrows might also need a legend.
Author Response
- The authors stated in the abstract that the tubulin code is a combination of PTMs and tubulins isoforms. I totally agree with this definition and I think that it represents the most commonly accepted and accurate definition to date. However, Table 1 and more generally the review are focused on tubulin PTMs only. There are several examples of links between tubulin isoforms expression (e.g. beta-III) and cancer prognosis or drug resistances (for example reviewed here doi: 3390/ijms18071434).
R: The manuscript and Table I now includes references to these studies and directs the reader to the indicated review for in depth coverage. We thank the reviewer for pointing it out.
- There are also experimental evidences that tubulin isoforms affect microtubule dynamic properties.
R: We now also include references to these works. Kapoor
- The review is still relevant and there is already a lot to mention only on PTMs but I think this point must be clarified either by mentioning the knowledge on tubulin isoforms (which might be too long and out of the scope of this review) or by assuming the focus on (some) PTMs and therefore only on a truncated version of the code. In that case, the words “the tubulin code” and especially “the cancer tubulin code” although very trendy might be misleading and overstated. As the authors said on the conclusion, a lots remains to be done to understand the “complete mitotic tubulin code” but I’m afraid that reducing the tubulin code to PTMs would be inaccurate. The cancer tubulin code (if there is one) can certainly not be reduced to PTMs. Maybe the use of “A code” instead of a “The code…” or any other way to clarify this statement will be fair and useful to not compromise the future use and the validity of the term “tubulin code” and thence, the value of this review.
R: We now cover also the most significant alteration of tubulin isotypes in the review and provide a balanced account of both alterations of tubulin isotypes and PTMs in cancer. We also removed the term “mitotic tubulin code” for accuracy.
- Line #60-61 the authors make the link between dynamic microtubules and tyrosination. Since the review might be addressed to readers not necessarily familiar with tubulin posttranslational modifications of tubulin and all the different reactions involved, it might be useful to explain slightly more this cycle. Reactions specificity on soluble versus polymeric tubulin and the link between dynamic microtubules and tyrosinated state might not be obvious for everyone.
R: These clarifications are now provided.
- Line #66, I see what the authors meant here with the difference between astral and spindle microtubules but the word “asymmetric” to describe the spindle could be confusing here since the two “half” spindles in the bipolar structure are symmetric (like in Figure 1).
R: We replaced the term “asymmetric” by “anisotropic”.
- Line #169, the mutation that compromise cytokinesis have been shown in the ciliate Tetrahymena which represents a particular context in terms of microtubule populations and post-translational modifications compare to other systems like mammalian cells. Although the interpretation is correct, the authors might want to state here that this phenotype was observed in a ciliate especially since the mutation in this context can also affect (poly)glycylation.
R: This is now clarified in the text.
- Line #190 when mentioning non-tubulin substrates of polyglutamylation and of TTLL4 enzyme, the reference [85] is relevant for cancer but the modification of non-tubulin substrates by TTLL4 was to my knowledge first reported in 2008 by van Dijk et al. 2008 JBC “Polyglutamylation Is a Post-translational Modification with a Broad Range of Substrates”. The modification of other substrates such as nucleosome assembly proteins (NAPs) was reported in 2000 by Regnard et al. 275, 15969 –15976
R: This is now corrected.
Comments on Figures
Figures and tables a simple and clear enough. However, given the quite large size of Figure 1, I think that an effort can be made to better visualize some details such as CENP-E motors that are really small. The legend for the orange dots (kinetochore) is missing and the different arrows might also need a legend.
R: This is now corrected as suggested.